# Umbilical Cord Tensile Strength Under Varying Strain Rates

**DOI:** 10.3390/bioengineering12080789

**Published:** 2025-07-22

**Authors:** Maria Antonietta Castaldi, Pietro Villa, Alfredo Castaldi, Salvatore Giovanni Castaldi

**Affiliations:** 1High Risk Pregnancy Unit, Azienda Ospedaliera Universitaria San Giovanni di Dio e Ruggi d’Aragona, 84131 Salerno, Italy; 2Mechanical Testing Department, RTM Breda, 84 Via Po, 20032 Cormano, Italy; pietro.villa@forgital.com; 3Industrial Planning and Development Department, Salerno Energia Distribuzione S.p.A., 84134 Salerno, Italy; alfredo.castaldi@grupposistemisalerno.it; 4Department of Transfusional Medicine, Azienda Ospedaliera Universitaria San Giovanni di Dio e Ruggi d’Aragona, 84131 Salerno, Italy; scastaldi@unisa.it

**Keywords:** umbilical cord, tensile strength, non-Newtonian behavior, mechanical test, biomechanical tissue properties

## Abstract

The tensile strength of the umbilical cord (UC) is influenced by its composition—including collagen, elastin, and hyaluronan—contributing to its unique biomechanical properties. This experimental in vitro study aimed to evaluate the UC’s mechanical behavior under varying strain rates and to characterize its viscoelastic response. Twenty-nine UC specimens, each 40 mm in length, were subjected to uniaxial tensile testing and randomly assigned to three traction speed groups: Group A (*n* = 10) at 8 mm/min, Group B (*n* = 7) at 12 mm/min, and Group C (*n* = 12) at 16 mm/min. Four different parameters were analyzed: the ultimate tensile strength and its corresponding elongation, the elastic modulus defined as the slope of the linear initial portion of the stress–strain plot, and the elongation at the end of the test (at break). While elongation and elongation at break did not differ significantly between groups (one-way ANOVA), Group C showed a significantly higher ultimate tensile strength (*p* = 0.047). A linear relationship was observed between test speed and stiffness (elastic modulus), with the following regression equation: y = 0.3078e^4.425x^. These findings confirm that the UC exhibits nonlinear viscoelastic properties and strain-rate-dependent stiffening, resembling non-Newtonian behavior. This novel insight may have clinical relevance during operative deliveries, where traction speed is often overlooked but may play a role in preserving cord integrity and improving neonatal outcomes.

## 1. Introduction

The umbilical cord (UC) is crucial in prenatal life, connecting the fetus to the placenta and thus providing an oxygenated blood supply together with nutrients [1]. Thus, to perfectly fulfill its duty, the UC structure is fundamental; it is composed of two arteries, containing blood derived from the fetus, and a vein, containing blood originating from the placenta, surrounded by Wharton’s jelly (WJ), a porous connective tissue, and an outer, single-cell layer of amnion [1,2].

The first description of WJ by Thomas Wharton dates back to 1656. It is a mucous connective tissue of the UC located between the amniotic epithelium and the umbilical vessels [2]. McElreavey et al. (1991) were the first to isolate mesenchymal stem cells (MSCs) from the WJ portion of the UC [3]. Several studies have indicated that WJ-MSCs can be used in many fields, such as neurological disorders [4], kidney injury [5], lung injury [6], orthopedic injury [7], liver injury [8], and cancer therapy [9].

Therefore, in obstetrical clinical practice, the mechanical properties of the UC, though often underestimated, are essential to maintain an unimpeded flow throughout pregnancy and labor [10]. This is primarily ensured by variations in mechanical properties, with the core being stiffer near the fetus compared with the peripheral WJ [10,11]. Moreover, the presence of Hyrtl’s anastomosis further protects the arterial components of the UC [12].

A limited number of studies reported on the biomechanical behavior of the UC and its components; nonetheless, a large amount of knowledge remains uninvestigated [10,11,13,14,15,16,17,18,19,20,21,22,23,24,25]. Experimental studies focusing on the mechanical behavior of the UC are mainly conducted with medicolegal purposes, where the strength of the UC is investigated in case of avulsion, questioning how much force is needed to result in a complete rupture of an UC or if it is possible that the UC will tear completely in case of precipitate delivery or operative delivery [14,16]. Indeed, when labor ends in an operative delivery, the cord is often not considered as a variable. Moreover, the impact of traction force on perinatal outcome after vacuum extraction is insufficiently studied. Although vacuum extraction is a common alternative to second-stage cesarean section (CS) [26,27,28,29], the method has not been developed substantially since its clinical implementation about 50 years ago, and the American College of Obstetricians and Gynecologists states that specific aspects of the technique have been poorly investigated [29]. Numerous studies on plastic cup vacuum extractions suggest that most deliveries require a traction force ranging between circa 110 and 250 Newton [26,27,28]. However, in these studies, the UC is again not considered a variable. In addition, the speed of traction is also not considered.

It has been previously reported that the UC exhibits both viscoelastic behavior, similarly to all biological tissues, and nonlinear mechanical properties [11,22]. Like other biological tissues, the UC can be viewed as a composite material reinforced with fibers. Specifically, the extracellular matrix (ECM) and myofibroblasts in Wharton’s jelly serve as the reinforcing fibers, while their orientation influences the anisotropy of the tissue’s mechanical behavior [22,30,31,32,33]. Additionally, hyaluronan, the primary component of the ground substance in the UC [34], exhibits pseudoplastic behavior, with its viscosity varying according to the shear rate [35].

Therefore, the aims of the present study were to evaluate the biomechanical behavior of the UC in different experimental conditions, to analyze biomechanical properties related to strain rate, and to examine the mechanical response of this unique tissue.

## 2. Methods

### 2.1. Data Collection and Patient Selection

The present experimental in vitro biomechanical study was performed at the Azienda Ospedaliera Universitaria San Giovanni di Dio e Ruggi d’Aragona Hospital, Salerno, Italy, and at the RTM Breda Laboratories, Cormano, MI, Italy. Our institutional review board approved this study (Ethics Committee “Campania Sud”, Brusciano, Naples, Italy; prot./SCCE n. 24988), which was carried out in accordance with the Code of Ethics of the World Medical Association (Declaration of Helsinki).

We collected UC samples from 5 healthy full-term pregnancies delivered via cesarean section. Written informed consent from the women involved in this study was obtained prior to sample collection. Demographic data, including age, height, weight, body mass index (BMI), previous surgery, parity, gestational age, and neonatal weight, were collected (Table 1).

The inclusion criteria were physiological pregnancies between 37 and 40 weeks, deliveries of neonates with birth weights from 2.5 to 4.0 kg, and UCs with normal lengths from 40 to 70 cm and with two arteries and one vein [36]. The exclusion criteria were cords of placentas with macroscopic evidence of inflammation, clinical or laboratory evidence of maternal infection (body temperature of >38°, heart rate of >100 bpm, respiratory rate of >24 breaths per minute, and white cell count of >15,000), and the hyper- or hypo-coiling of the cord. UCs were stored in physiological saline immediately after delivery to preserve the mechanical properties and avoid dehydration, and they were stored at −20 °C. The testing was performed at room temperature (23–32 °C), with 40–60% humidity [37,38], mimicking the conditions of a delivery room.

### 2.2. Biomechanical Tests

A total of 29 specimens of UCs, which were obtained from the five collected UCs and cut to measure 50 mm in length, underwent stress–strain testing. Each specimen, after the verification of its integrity, was randomly assigned to 1 of 3 different speed traction groups: Group A (*n* = 10) with an 8 mm/min traction speed; Group B (*n* = 7) with a 12 mm/min traction speed; and Group C (*n* = 12) with a 16 mm/min traction speed.

The specimens were mounted onto pneumatic jaws by interposing a brass sheet lined with gauze between the biological material and the friction surface in order to increase the adhesion of the sample and decrease the edge effect, thus preventing both slippage and rupture during tensile testing. Once secured in the pneumatic jaws, the sample had a final length of 40 mm, with 5 mm held between the jaws. Each clamp was tightened enough to hold the specimen in place during testing, but not enough that failure would occur at the interface with the clamp and specimen.

The machine used, an MTS Alliance 10 tensile tester (matr. 00782, MTS System Corporation, 14000 Technology Drive, Eden Prairie, MN, USA), was equipped with an additional AEP load cell with 3500 N at the full scale. The bottom clamp was attached to the immobile platform of the material testing machine, while the top clamp was attached to the mobile actuator of the machine (Figure 1).

Each specimen was first preloaded to 1 N by slightly raising the crosshead. The preload was chosen to be small enough so that the specimens would be properly tensioned and oriented. Using the sample free length (40 mm), the conventional strain rate (hereafter referred to simply as the “strain rate”) was defined for each test speed: Group A = 0.2 1/min, Group B = 0.3 1/min, and Group C = 0.4 1/min, respectively.

The tests were performed at room temperature, and the specimens were kept moist throughout with normal saline solution. The force and displacement signals from the test machine were captured with a sampling frequency of 10 Hz. Appendix A shows a single stress–strain test.

Four different parameters were retrieved from our biomechanical tests: the ultimate tensile strength and its corresponding elongation, the elastic modulus defined as the slope of the linear initial portion of stress–strain plot, and the elongation at the end of the test (at break) (Appendix A). All elongations were normalized by the sample standard free length (40 mm) to obtain the dimensionless strain. The recorded forces were converted into normal stress using a conventional cross-section area of 177 mm^2^, corresponding to a mean cord diameter of 15 mm.

The raw data are available in Appendix A.

### 2.3. Statistical Analysis

The primary goal of this study was to evaluate the ultimate tensile strength (UTS) and the corresponding cord elongation (Elo@UTS). Once the test was completed, the resulting curve was processed to identify its initial linear portion, which was used to determine the elastic modulus (E) and the cord elongation at break (Elo@Break).

A power calculation was undertaken to determine an appropriate sample size for this study. We calculated the mean and standard deviation of the cord length and elongation, assuming that in an operative delivery, at each traction, a descent of the fetal head of 30% is expected [26,28]. Thus, we used a mean of 40 mm for the length of the cord on the basis of a recent study that states that this is the minimum size to have a laminar flow in the cord, with a standard deviation of 10 mm [39].

A two-sided test power calculation was performed in order to detect the maximum cord elongation. Starting from a baseline of 40 mm, we hypothesized 30% elongation until break. The standard deviation was set to 10 mm. This power calculation indicated that 7 specimens in each group would be necessary to detect a 15% difference in the maximum cord elongation with a power of 80% at a 5% level of significance.

The secondary endpoint was to evaluate the dependency between the sample’s stiffness and the speed of the test.

The data distribution was assessed using the Shapiro–Wilk test and showed a parametric distribution. No patients had missing data. Data were analyzed by using one-way ANOVA. Statistical calculations were performed using the Statistical Package for the Social Sciences (SPSS) software (version 29.0; SPSS Inc., Chicago, IL, USA) and Microsoft Office 365 (2023; Microsoft Corporation, Redmond, WA, USA). A value of *p* < 0.05 was considered statistically significant. Given that the group degrees of freedom (df) were 2, the critical F-value was 3.369.

## 3. Results

Overall, the analyzed parameters tended to increase with the strain rate; however, the one-way ANOVA test revealed a statistically significant relationship only for the elastic modulus and UTS values (Table 2).

In terms of UTS, Group A (0.2 1/min) recorded a mean value of 22.2 N/cm^2^ with a standard deviation (SD) of 11.8. Group B (0.3 1/min) showed a slightly lower mean UTS of 21.3 N/cm^2^ (SD = 6.2), while Group C (0.4 1/min) exhibited the highest mean UTS at 32.6 N/cm^2^ (SD = 12.0) (*p* = 0.047; F = 3.439).

E followed a similar trend. Group A demonstrated the lowest stiffness, with a mean value of 0.84 MPa (SD = 0.56), followed by Group B with 1.29 MPa (SD = 0.34), while Group C reached the highest mean value of 1.97 MPa (SD = 0.96) (*p* = 0.004; F = 6.800).

With regard to Elo@UTS, Group A reached a mean of 34% (SD = 23), while Group B showed a notably lower mean elongation of 17% (SD = 6). Group C had an intermediate mean value of 22% (SD = 15) (*p* = 0.103; F = 2.483).

Finally, Elo@Break was higher in Group A, with a mean of 55% (SD = 26), compared with 35% (SD = 12) in Group B and 36% (SD = 20) in Group C (*p* = 0.066; F = 3.032).

Finally, there was an exponential dependency between the average UTS and the strain rate $\dot{\varepsilon }$ (Figure 2), as well as between the average modulus of elasticity E and the strain rate $\dot{\varepsilon }$ (Figure 3).

Representative stress–strain curves corresponding to the three analyzed strain rates are shown in Appendix A.

## 4. Discussion

The biomechanical structure of the UC is a relatively underexplored topic, especially regarding its implications in the delivery room setting [26,27,28,29]. However, in recent decades, medical and popular interest in UC-related issues has increased. Notably, ten years ago, J.H. Collins published the Second Edition of *Silent Risk: Issues About the Human Umbilical Cord*, which comprehensively addressed recent findings and highlighted the UC as an overlooked fetal adnex [40].

Consistent with the previous literature reporting UC elongation ranging from 12% to 70% [22,23,24,25,40], our findings show mean elongation values between 35% and 55%. Importantly, two novel observations emerged from our data. First, the maximum peak load was recorded at the highest displacement speed. Second, the acquired data show that the elastic modulus increased with the applied strain rate, a behavior that is characteristic of amorphous polymeric materials (Table 2). This trend aligns with the viscoelastic nature of the UC, which exhibits both instantaneous elastic and time-dependent viscoelastic deformation responses [41,42,43,44].

This supports our hypothesis that the UC behaves similarly to a polymeric chain at varying strain rates, displaying the characteristics of non-Newtonian fluids. The observed strain-rate-dependent behavior supports the hypothesis that the UC behaves similarly to polymeric chains and non-Newtonian fluids. Previous research on polymers such as polymethylmethacrylate (PMMA) and polyamideimide (PAI) shows that their mechanical behavior can vary significantly across different temperature and strain rate conditions, transitioning from rubbery to ductile plastic and, ultimately, to brittle states [41].

Extensive studies on PMMA and PAI have provided valuable insights in this regard. PMMA is a widely used commercial polymer that is appreciated for its optical clarity and mechanical resilience, particularly in high-stress environments such as impact-resistant aircraft windows. Similarly, PAI is a high-performance polymer engineered for stability under extreme temperature and mechanical loads, including applications in aerospace engineering, such as components of space shuttles [41]. In both PMMA and PAI, increased strain rates have been shown to significantly elevate the initial Young modulus due to the reduced molecular mobility and increased stiffness of polymer chains under rapid deformation [44]. Comparable mechanical behavior can be hypothesized for the UC, particularly when considered as a fiber-reinforced composite material [30,31,42].

Indeed, biologically, the UC can be modeled as a collagen fiber-reinforced composite, where a solid matrix supports mechanical loads [30,31,40]. According to Holzapfel et al. [42], the strain energy function (SEF) of the isotropic non-collagenous matrix (e.g., elastin fibers, cells, ground substances—distinct from the ECM) differs significantly from the anisotropic SEF of the collagen network. While Wharton’s jelly (WJ) contains ~50% collagen, over 70% of its glycosaminoglycan (GAG) content is hyaluronan, a molecule with pronounced non-Newtonian properties due to its shear-rate-dependent viscosity [34,35,43].

Therefore, several constitutive models have been proposed in the technical literature to characterize the viscoelastic behavior of solid materials, which encompasses both an instantaneous elastic response and a time-dependent (viscous) deformation component (as illustrated in Appendix A) [45,46]. This stress–strain relationship is governed by the applied strain rate, which, as evidenced by our findings, can significantly influence the mechanical performance of the umbilical cord (UC).

Basic viscoelastic behavior can be modeled using the Maxwell model, which consists of a spring and dashpot in series, effectively capturing stress relaxation under constant strain [43]. Conversely, the Kelvin–Voigt model, comprising a spring and dashpot in parallel, is commonly used to represent creep behavior under constant stress [46]. A more comprehensive representation of viscoelastic properties, particularly those exhibiting both relaxation and creep behaviors, is given in the Standard Linear Solid (SLS) model. This model integrates a spring in parallel with a Maxwell element (i.e., a spring and dashpot in series), offering a more accurate approximation of biological tissue mechanics under dynamic loading conditions [47]. The analytical relationship between stress and strain provided by our results can be explained by the following equation (Figure 4a).

In this equation, σ is the applied stress, E and E1 are the stiffness of the material, ε is the strain, and η is the viscosity of the dashpot component (damper). The functional scheme of the SLS model is depicted in Figure 4b. This functional scheme of the SLS model is given by a parallel with a spring (lower arm in picture) and a Maxwell element (upper arm), composed of a spring in series with a dashpot. In the case of a slowly applied load, both arms react, contributing to the total deformation, and the strain is the sum of the elastic and viscoelastic components. Conversely, in the case of the sudden application of the force, the damper increases its stiffness, and the sample’s strain is predominantly governed by the lower arm. In other words, the lower arm represents the pure elastic response, while the upper arm shows the viscoelastic response, which is consistent with the mechanical behavior observed in biological tissues such as the umbilical cord.

Thus, this study represents the first contribution to the biomechanical literature that characterizes the strain-rate-dependent viscoelastic behavior of the UC, particularly emphasizing its strain hardening response. This phenomenon, whereby stiffness increases with strain rate, may have significant clinical implications, especially in the context of operative deliveries such as vacuum extraction. Notably, current clinical protocols often overlook traction velocity as a critical parameter, despite our findings suggesting it could serve as a cord-preserving and potentially life-saving factor. Indeed, the results of our study suggest that traction speed may play a significant role in preserving the structural integrity of the cord. By minimizing mechanical stress and potential damage, modulating traction velocity during operative deliveries could enhance both delivery safety and neonatal outcomes.

We propose that the observed rate-dependent increase in stiffness could be incorporated into future clinical investigations involving instrumental deliveries. Our data (Figure 3) demonstrate a linear relationship between the strain rate and elastic modulus, supporting the hypothesis that the UC exhibits typical features of amorphous, viscoelastic polymeric materials under dynamic loading.

Finally, these preliminary findings, while promising, require further validation with larger sample sizes and a broader range of strain rates. Indeed, the limited sample size and the restricted range of applied strain rates constrain the generalizability of the results. Future research should also explore three key directions. First, time-dependent biomechanical studies are needed to analyze the deformation responses of this biological tissue to enhance the accuracy of biomechanical models. Therefore, further research involving a larger sample and an extended set of traction velocities is necessary to confirm and expand upon these observations, ensuring their robustness and applicability in clinical settings. Second, computational modeling, by integrating the observed mechanical parameters into finite element or other predictive models, could be used to simulate UC behavior under varying loading conditions. Third, the viscoelastic properties of WJ should be leveraged to develop biocompatible scaffolds for use in regenerative medicine and tissue engineering.

## Figures and Tables

**Figure 1 bioengineering-12-00789-f001:**
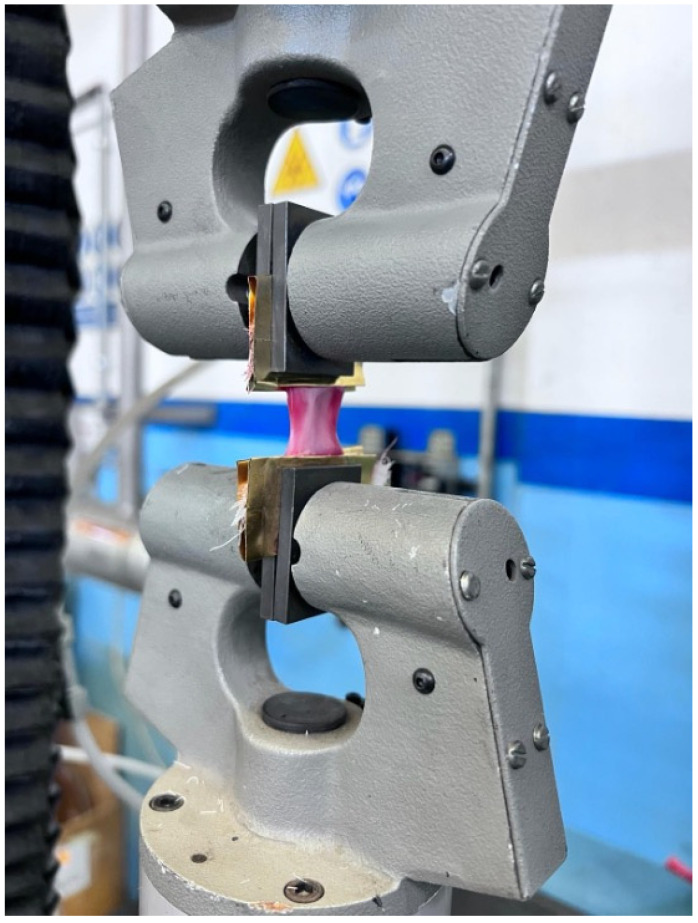
An umbilical cord specimen on the MTS Alliance 10 tensile tester (matr. 00782, MTS System Corporation, 14000 Technology Drive Eden Prairie MN, USA).

**Figure 2 bioengineering-12-00789-f002:**
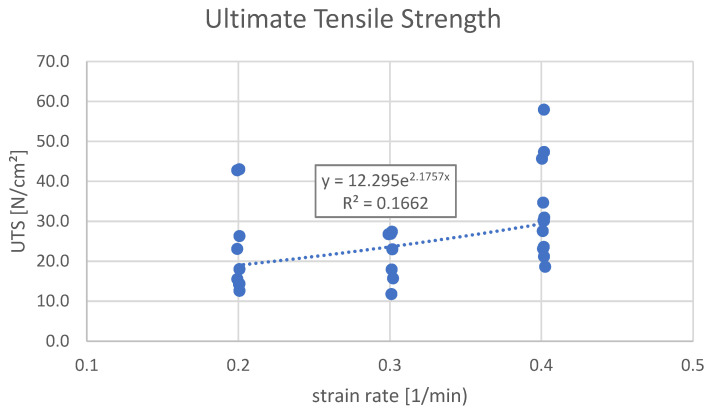
Ultimate tensile strength vs. strain rates. Group A: 0.2 at strain rate of 1/min; Group B: 0.3 at strain rate of 1/min; Group C: 0.4 at strain rate of 1/min. There is an exponential dependency between the average UTS and the strain rate $\dot{\varepsilon }$, which is described by the following equation: UTS_AVG_ [N/cm^2^] = 12.295·EXP(2.1757 $\dot{\varepsilon }$ [1/min]).

**Figure 3 bioengineering-12-00789-f003:**
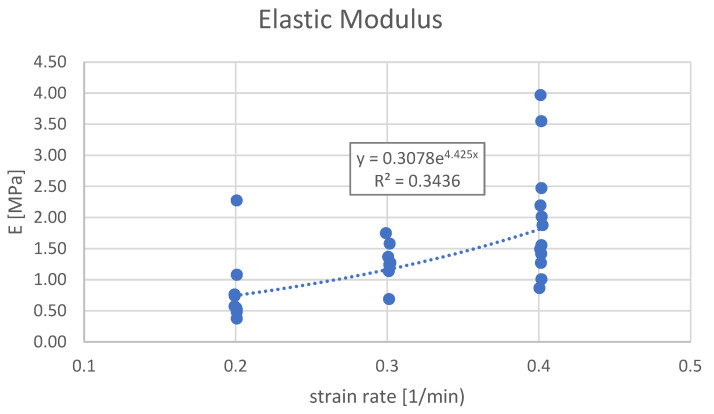
Elastic modulus vs. strain rates. Group A: 0.2 at strain rate of 1/min; Group B: 0.3 at strain rate of 1/min; Group C: 0.4 at strain rate of 1/min. There is an exponential dependency between the average modulus of elasticity E and the strain rate $\dot{\varepsilon }$, which is described by the following equation: E_AVG_ [MPa] = 0.3078·EXP(4.425 $\dot{\varepsilon }$ [1/min]). Average stiffnesses (black dots) vs. applied strain rates. A = Group A, 8 mm/min; B = Group B, 12 mm/min; C = Group C, 16 mm/min. There is a linear dependency between the samples’ stiffness (modulus of elasticity) and the speed of the test (strain rate normalized per initial length of the samples), as shown by the equation.

**Figure 4 bioengineering-12-00789-f004:**
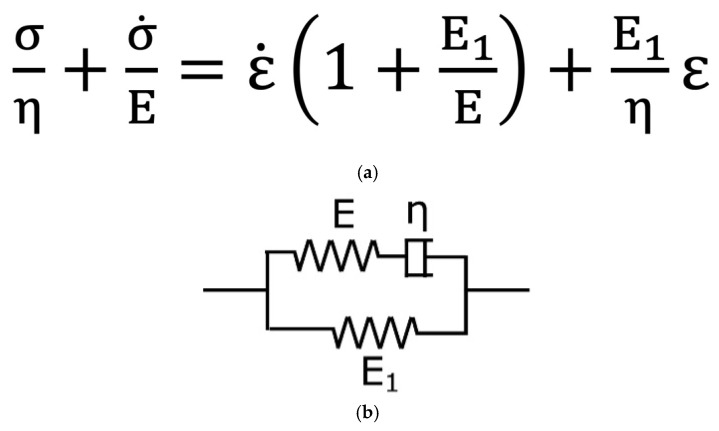
(**a**) An equation representing the analytical relationship between stress and strain provided by the Standard Linear Solid (SLS) functional model: σ is the applied stress, E and E1 are the stiffness of the material, ε is the strain, and η is viscosity of the dashpot component (damper). (**b**) The Maxwell representation of the SLS model consists of two systems in parallel, containing springs (E, E1) and a damper (η).

**Table 1 bioengineering-12-00789-t001:** Demographic and clinical characteristics of the patients whose umbilical cords were collected for the present study.

	(*n* = 5)
Age	36 ± 1.2 [32–39]
Height (m)	1.62 ± 0.02 [1.56–1.67]
Weight (kg)	91.8 ± 5.5 [76.4–107.1]
BMI (kg/m^2^)	34.8 ± 1.5 [30.7–38.8]
Previous surgery	40%
Parity	0.8 ± 0.3 [0–1.8]
Gestational age	37 ± 4 [35.5–38.1]
Neonatal weight (grams)	3106 ± 140 [2716–3495]

Continuous variables are expressed as means ± SE with 95% CIs, whereas discrete variables are expressed as percentages.

**Table 2 bioengineering-12-00789-t002:** Results of the biomechanical tests of the 29 specimens.

Biomechanical Parameters §	Group A (0.2 1/min) (*n* = 10)	Group B (0.3 1/min) (*n* = 7)	Group C (0.4 1/min) (*n* = 12)	ANOVA Values	
				*p* (*)	F (**)
Strain Rate, 1/min	0.20	0.30	0.40	-	-
Ultimate Tensile Strength (UTS) vs. Strain Rate, N/cm^2^				0.047	3.439
*AVG value*	22.2	21.3	32.6		
*MIN, MAX values*	12.6, 43.0	11.8, 27.4	18.6, 57.9		
Elongation at UTS (Elo@UTS) vs. Strain Rate, %				0.103	2.483
*AVG value*	34	17	22		
*MIN, MAX values*	12, 89	10, 28	5, 58		
Elongation at Break (Elo@Break) vs. Strain Rate, %				0.066	3.032
*AVG value*	55	35	36		
*MIN, MAX values*	23, 103	21, 55	11, 76		
Elastic Modulus (E) vs. Strain Rate, MPa				0.004	6.800
*AVG value*	0.84	1.29	1.97		
*MIN, MAX values*	0.37, 2.287	0.69, 1.75	0.86, 3.97		

* *p*-value < 0.05 indicates correlation between two factors. ** F < 3.369 (2 df) indicates that the test is statistically significant (confidence level: 95%). § Biomechanical parameters: UTS (N/cm^2^; maximum force recorded/cross-section); Elo@UTS (% of length at maximum load/initial length); Elo@Break (% of length at rupture/initial length); E (MPa; slope of initial linear portion of stress/strain plot).

## Data Availability

The data presented in this study are available as Appendix A.

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
