# Peer review of "Umbilical Cord Tensile Strength Under Varying Strain Rates"

_bioengineering, 2025, doi:10.3390/bioengineering12080789_

Round 1
Reviewer 1 Report
Comments and Suggestions for Authors
Thank you for the invitation to review the article.
The manuscript is well-written but I believe that some points can be improved. The study design, inclusion/exclusion criteria, sample size, and statistical techniques need to be described with sufficient clarity to allow replication, including quality control and any changes during data collection.
As for the discussion, it should directly relate the results obtained to the proposed objectives, contextualize with the existing literature (including conflicting works), and transparently address limitations, biases, and practical implications/relevance.
Comments on the Quality of English LanguageAs for the quality of English, I do not have the technical qualifications to comment on this issue.
Author Response
The manuscript is well-written but I believe that some points can be improved.
We thank the reviewer for this favorable comment.
The study design, inclusion/exclusion criteria, sample size, and statistical techniques need to be described with sufficient clarity to allow replication, including quality control and any changes during data collection.
We thank the reviewer for these constructive comments. This study is an experimental in vitro investigation, which has been more clearly specified (highlighted in green) both in the abstract (page 2) and in the Methods section (page 5). The inclusion and exclusion criteria, as well as the sample size, have been further detailed in the Methods section (page 5). The description of the statistical analysis has also been refined (Methods section, pages 6 and 7). Each sample underwent a quality control check to verify its integrity, and this procedure has been added to the Methods section (page 7). No changes were made during data collection.
As for the discussion, it should directly relate the results obtained to the proposed objectives, contextualize with the existing literature (including conflicting works), and transparently address limitations, biases, and practical implications/relevance.
We thank the reviewer for these insightful suggestions. We have emphasized more clearly that this study represents the first experimental investigation of the biomechanical response of human umbilical cord tissue under varying traction speeds. Additionally, we have further highlighted the clinical significance and potential practical applications of our findings. Moreover, we have clarified the limitations of the present study in accordance with the reviewer’s recommendations (revisions have been highlighted in green in the Discussion section, page 11).
Reviewer 2 Report
Comments and Suggestions for Authors
The authors evaluated the tensile strength of the umbilical cord at three different velocities, and this is the third revision of the manuscript. I believe that the authors provided a comprehensive and appropriate answer, and the manuscript is now without any faults. The interpretation of the results and the discussion is adequate, and the research area is underscored, but I think it is rather important. The study revealed the non-linear viscoelastic properties and non-Newtonian behaviour of the UC, however, a linear relationship was observed between test speed and stiffness. I accept the MS in its present form. I could not add anything to the MS.
Author Response
The authors evaluated the tensile strength of the umbilical cord at three different velocities, and this is the third revision of the manuscript. I believe that the authors provided a comprehensive and appropriate answer, and the manuscript is now without any faults. The interpretation of the results and the discussion is adequate, and the research area is underscored, but I think it is rather important. The study revealed the non-linear viscoelastic properties and non-Newtonian behaviour of the UC, however, a linear relationship was observed between test speed and stiffness. I accept the MS in its present form. I could not add anything to the MS.
We thank the reviewer for these favorable comments.